# Superiority of Cellulose Non-Solvent Chemical Modification over Solvent-Involving Treatment: Solution for Green Chemistry (Part I)

**DOI:** 10.3390/ma13112552

**Published:** 2020-06-03

**Authors:** Stefan Cichosz, Anna Masek

**Affiliations:** Faculty of Chemistry, Institute of Polymer and Dye Technology, Lodz University of Technology, 90-924 Lodz, Stefanowskiego 12/16, Poland; stefan.cichosz@dokt.p.lodz.pl

**Keywords:** cellulose, non-solvent modification, silane, green chemistry

## Abstract

In the following article, a new approach of cellulose modification, which does not incorporate any solvents (NS), is introduced. It is compared for the first time with the traditional solvent-involving (S) treatment. The analysed non-solvent modification process is carried out in a planetary mill. This provides the opportunity for cellulose mechanical degradation, decreasing its size, simultaneously with ongoing silane coupling agent grafting. Fourier-transform infrared spectroscopy (FT-IR) indicated the possibility of intense cleavage of the glucose rings in the cellulose chains during the mechano-chemical treatment. This effect was proved with dynamic light scattering (DLS) results—the size of the particles decreased. Moreover, according to differential scanning calorimetry (DSC) investigation, modified samples exhibited decreased moisture content and a drop in the adsorbed water evaporation temperature. The performed research proved the superiority of the mechano-chemical treatment over regular chemical modification. The one-pot bio-filler modification approach, as a solution fulfilling green chemistry requirements, as well as compromising the sustainable development rules, was presented. Furthermore, this research may contribute significantly to the elimination of toxic solvents from cellulose modification processes.

## 1. Introduction

At the present time, substances provided from natural resources, e.g., plant-based fibres [1,2,3], aluminosilicates [4,5,6], polyphenols [7,8], are crucial when considering the opportunity of bio-based polymer composite production [9,10,11,12]. This is reflected in an increasing interest in research areas and manufacturing concerning eco-friendly materials [13]. Among the mentioned above substances, natural fillers, because of their ease of processing, low cost, low density, biodegradability and good mechanical properties, have been widely investigated [14].

Nowadays, plant-based fibre reinforced polymer composites are widely used as low-cost and sustainable materials in structural or non-structural applications. Unfortunately, they do not provide the same level of reinforcement compared to the incorporation of carbon black or silica [15]. The main problem is the poor interface between the hydrophilic bio-filler and the non-polar polymer matrix [16]. Consequently, the surface hydrophobisation of cellulose fibres is highly recommended in order to obtain a material of sufficient mechanical and thermal properties [17,18].

There are known different methods of natural fibre modification, e.g., employing maleinized polymer matrix as a compatibilizer [19,20]. Yet, cellulose is often modified with silanes as they are substances of a high efficiency considering the hydrophobisation of natural fibre surfaces [2,21,22]. Moreover, the treatment procedure is not complicated. Silanes may affect different properties of cellulose, e.g., moisture content, thermal decomposition, surface characteristics, fibre size [23,24,25,26,27]. Some natural-based fibre silane treatments and their effect on the biopolymer properties are described below.

Thakur et al. [23] proposed modification of Eulaliopsis binata fibres preceded by mercerization in 2% NaOH solution and then treatment with vinyltrimethoxysilane (VTMS) in ethanol/water (60/40) mixture; VTMS concentration—5%, pH = 3.5–4, 2h. Due to the blockade of active functional groups (hydroxyl moieties), a decrease of water uptake and an increase in chemical resistance could be observed. Moreover, silanization contributed to some slight variations in the cellulose initial decomposition temperature and elevated the final decomposition temperature by 10 °C.

Furthermore, Ramamoorthy et al. [24] treated lyocell fibres in an alkali environment for different times (0.5–72 h) and at different temperatures (25–50 °C). Then the biopolymer was modified with 3-aminopropyl-triethoxysilane (APTES). Treatments contributed to weight loss, changes in mechanical properties, and improvement of thermal resistance. The chemical analysis revealed that the weight loss was due to the decrease of the hemicellulose content (α-cellulose content remained constant). Removal of hemicellulose caused void creation within the fibre structure and, therefore, the mechanical properties of the filler were altered. Moreover, the impact of alkali treatment on moisture content decrease was found to be more significant in comparison with the silanization process as Si–O–cellulose bonds are not stable against moisture. Yet, for silane treatment, water absorption was lower due to the surface hydrophobisation and blockade of –OH moieties.

Moreover, Kabir et al. [25] referred that alkali treatment (0–10% NaOH in water/ethanol mixture) of hemp fibres caused an efficient purification of the natural filler as the hemicellulose and lignin content decreased. Then, such prepared fibres were modified with an oligomeric siloxane (3%) dissolved in a methyl alcohol solution. This contributed to further purification and improvement in thermal resistance. Silane treatment was found to form a covering on the fibre surface and to fill the voids between the microfibrils.

On the other hand, Nakatani et al. [26] also investigated the reactivity of different silanes with fibrous cellulose in order to create a syndiotactic polypropylene (sPP) composite sample: aminopropyltrimethoxysilane (APTMS), aminopropyltriethoxysilane (APTES), isobutyltrimethoxysilane (IBTMS). The highest reactivity with the cellulose hydroxyl groups was found for APTMS. Therefore, it was claimed that the linear silane compound with methoxyl groups is the most suitable for the reaction.

Concerning the information gathered above, it could be observed that the silanization of natural fibres has been widely investigated in recent years. Unfortunately, most of the modification processes carried out on the natural fillers involve solvents in the treatment procedure. It should be emphasized that their employment in the modification process, undoubtedly, contributes to increased toxicity, hazards, pollution, and an increase of waste [28].

This goes against the rules of green chemistry and challenges the idea behind the creation of an eco-friendly alternative to commonly used polymeric materials because the process of their production should be non-toxic as well [29].

Yet, it should be considered that solvents play an important role in the treatment procedure. They create a certain environment in which the chemical reaction occurs. Therefore, solvents are crucial regarding the mass/heat transfer, reaction rate, selectivity, or position of chemical equilibrium [28]. Taking into consideration the given information, it is essential to invent an eco-friendly modification process of plant-based fibres which would compromise the sustainable development rules, as well as provide the desired reaction yield and time.

Consequently, this article describes the solvent-free mechano-chemical approach of cellulose fibre modification and for the first time compares the obtained results with the traditional solvent-involving treatment. Here, milling of natural fibres is carried out instead of a mercerization process and the further silanization is performed in the same vessel.

The aim of mercerization is to alter the cellulose for further chemical modification and the milling process is similar in principle. Yet, solvent is not required. Therefore, in the following article an innovative one-pot method for bio-filler hydrophobisation, not compromising the rules of green chemistry, is presented. The new approach of cellulose modification may contribute to the elimination of various toxic solvents from currently performed natural fibre treatments.

## 2. Materials and Methods

### 2.1. Materials

The Arbocel^®^ UFC100 Ultrafine Cellulose for Paper and Board Coating (UFC100) was purchased from J.Rettenmaier & Söhne (Rosenberg, Germany). The mentioned material is in a powder form (white, odourless) and its density equals approximately 1.3 g/cm^3^. Cellulose is hard to dissolve in water, fats, and many commonly used solvents. On the other hand, UFC100 may bind water to its surface very easily. The length of the fibres varies between 6–12 μm and the pH is from 5–7.5. The cellulose moisture content is approximately 6 wt% (estimated on the basis of the previous study [30]).

Cellulose fibres were treated with three types of silane: (i) triethoxyoctylsilane (TEOS) from Sigma-Aldrich—a transparent and colourless liquid, soluble in water, melting temperature about −75°C, boiling temperature approximately 85 °C; (ii) trimethoxypropylsilane (TMPS) from Sigma-Aldrich—a transparent and colourless liquid, soluble in water, boiling temperature approximately 142 °C; (iii) vinyltrimethoxysilane (VTMS) U-611 from UniSil—a transparent, colourless liquid, soluble in benzene, carbon tetrachloride and acetone, capable of reacting with water, melting temperature about −97 °C, boiling temperature approximately 123 °C. The chemical structures of silane coupling agents are presented in Figure 1.

Ethanol (concentration: 96%) employed as a reaction environment in the solvent-involving modification approach was purchased from Chempur (viscosity at 20 °C of 1.078 mPas, density at 20 °C about 0.79 g/cm^3^, vapor pressure at 20 °C at the level of 233 mbar, liquid soluble in water). All reagents were commercial products of the highest purity available.

### 2.2. Modification of Cellulose Fibres

In Table 1 the summary of all performed modification processes and sample name abbreviations are listed.

#### 2.2.1. Solvent-Involving Approach

In this kind of modification ethanol was employed as a reaction medium (ethanol [ml] to cellulose [g] ratio—20:1). Cellulose and silane (cellulose [g] to silane [ml] ratio—3:1) were stirred, with the use of a rotary evaporator in the presence of NH_3_^•^H_2_O (cellulose [g] to NH_3_^•^H_2_O [ml] ratio—15:2), in a flask for 2 h at 60 r/min (oil bath, 40 °C). The next step was to carry out the, the vacuum distillation (oil bath 60 °C, initial pressure 200 mbar). Solvent was removed and the specimens were dried in an oven for 4 h at 120 °C and further for 24 h at 100 °C. Then, the samples were stored in a dryer at 40 °C. Figure 2 reveals the scheme of cellulose silane treatment divided into the successive steps.

#### 2.2.2. Non-Solvent Approach

This type of modification was carried out in a planetary mill (Pulverisette 5, Fritsch, Merazet, Poznan, Poland). Cellulose and silane (cellulose [g] to silane [ml] ratio—3:1) were put into two steel containers (10 steel milling balls of 5 mm diameter in each vessel). The planetary mill was set for 2 h and 300 rpm. Then, the samples were dried in an oven for 4 h at 120 °C and further for 24 h at 100 °C. Then, the samples were stored in a dryer at 40 °C.

### 2.3. Characterization of Cellulose Fibres

#### 2.3.1. Fourier-Transform Infrared Spectroscopy (FT-IR)

Fourier transform infrared spectroscopy (FT-IR) measurement was carried out in the range from 4000–400 cm^−1^ (64 scans, absorption mode). Thermo Scientific Nicolet 6700 FT-IR spectrometer equipped with diamond Smart Orbit ATR sampling accessory was employed in this research. Before carrying out the experiment, cellulose fibres were stored for 24 h at 100 °C (laboratory oven, Binder, Tuttlingen, Germany).

#### 2.3.2. Dynamic Light Scattering (DLS)

The hydrodynamic radius of cellulose particles dispersed in the aqueous environment (0.1 g of the powder per 200 ml of distilled water) was determined with the dynamic light scattering technique. Solutions were subjected to ultrasound for 30 min. Only then, were the dispersion specimens placed in colorimetric cuvettes. The device employed in this research was a ZetaSizer Nano–S90 from Malvern Instruments (Malvern, UK).

#### 2.3.3. Differential Scanning Calorimetry (DSC)

Cellulose fibres were dried for 24 h at 100 °C (Binder oven) before being analysed. Differential scanning calorimetry (DSC) (Mettler Toledo, Greifensee, Switzerland) measurement was carried out in the temperature range from −20 to 200 °C (heating rate: 10 °C/min; Ar 60 cm^3^/min). Moreover, the water enthalpy (ΔH) and temperature of the peak (T_peak_) for the water evaporation process were determined. In the investigation the following device was employed: Mettler Toledo TGA/DSC 1 STARe System equipped with Gas Controller GC10.

## 3. Results and Discussion

### 3.1. Fourier-Transform Infrared Spectra Investigation

Unmodified and modified cellulose fibres were analysed using spectroscopic methods. The experiment was carried out not only in order to observe the changes in the UFC100 structure caused by the silanization process, but also to detect the variations of the impact on cellulose between the solvent-involving and non-solvent modification processes.

FT-IR spectra of UFC100 (Figure 3c–f) reveals some particular absorption bands characteristic of cellulose fibres, e.g., hydroxyl moieties at 3334 cm^−1^ [32] and 1030 cm^−1^ [33], C–H stretching vibration at 2896 cm^−1^ [34], –COO at 1200–900 cm^−1^ [35] which were also detected in different works [36,37]. Absorption bands assigned to the exact moieties are presented in Table 2.

Generally, the modification of cellulose with the use of the silane coupling agent is troublesome to observe in the FT-IR spectra due to the overlapping effect of the absorption bands assigned to the cellulose fibres and the signals originating from the silicon atom containing chemical groups. Therefore, it is hard to distinguish some changes in the intensities of given absorption bands. All analysed spectra revealed in Figure 3c–f have the same shape.

Only UFC100/TEOS/S exhibit some differences in comparison with the reference sample. Nevertheless, it is only a question of the whole spectrum intensity—it is lower. Repeated FT-IR measurements resulted in obtaining a spectra of the same intensity.

The peaks which are assigned to the C–O, C–C stretching vibrations and CH_2_ rocking vibration at 1100–900 cm^−1^ [35,43] correspond also to the Si–O–Si (1100 cm^−1^, 950–800 cm^−1^), Si–O–C (1450 cm^−1^), Si–OH (900 cm^−1^), and C–Si–C bonds (663 cm^−1^) [23,39,42]. This phenomenon for the silanization of the cellulose fibres has been observed in former studies [47]. Figure 3c,e reveals how difficult it is to evidence specific variations to confirm silane grafting. Changes in the intensities of these peaks may indicate some information about, e.g., cellulose degradation, as well as they might reveal silane coupling agent grafting on the fibre surface. Nevertheless, taking into consideration that these two processes are ongoing simultaneously, it is impossible to distinguish the intensity variations.

Likewise, new absorption bands do not appear in the FT-IR spectra, as silane coupling agents employed in this research are functionalised with alkyl chains of different length and structure—new functional groups are not introduced into the system; only silicon atoms containing moieties and carbon-based chains. Consequently, one should consider some subtle variations in the FT-IR spectra in order to confirm the modification process occurrence. As it is impossible to distinguish the intensity variations, it is advised to analyse the shifts between the absorption bands which, according to some researchers, is a more accurate method [46,48,49].

Interestingly, regarding the main absorption bands visible in Figure 3, shifts between the peaks could be detected. It is crucial considering the possibility of different interaction characteristics among cellulose fibres caused by some structural variations. Visible changes are stronger in the case of non-solvent treatment. This indicates that the fibres modified with this method are altered more during the conducted process. Many researchers explain absorption band shifts as factors indicating significant structural changes within cellulosic materials. Furthermore, the shifts are considered to be more accurate than peak intensity changes [46,48,49,50,51].

Moreover, according to the literature, some shifts visible at approximately 1030 cm^−1^ (glycoside bond asymmetric stretching vibration) may evidence degradation of glucose rings which could result in the particle size decrease of modified cellulose fibres [52]. Therefore, these shifts were evidenced more often for the non-solvent UFC100 treatment carried out in a planetary mill.

Furthermore, some variations in the intensity of the absorption band at 3330 cm^−1^ could be detected. This may indicate some information about blocking of –OH moieties of the cellulose chains by silanes coupled to the UFC100 surface. Nevertheless, these variations also may be explained by different moisture contents of the cellulose fibres which could be the outcome of the surface hydrophobisation occurring during the modification process.

On the basis of gathered results, it may be concluded that during the silanization of cellulose fibre degradation of glucose rings and surface hydrophobisation occurs. What is more, the modification with silane coupling agents may be confirmed by interaction changes among UFC100 samples which is evidenced by some strong shifts between the main absorption bands.

The modification process was also tracked with the employment of Near infrared spectoscopy (NIR) technique, which according to the literure, is more sensitive for polar groups [53,54,55]. Nevertheless, this method is not widely-known, but it seems a perfect tool for the cellulose structure [56,57] and moisture content [58,59,60,61] investigation. The NIR analysis is presented in the Appendix A.

### 3.2. Dynamic Light Scattering Investigation

Many cellulose treatments reported in the literature are associated with mass loss and particle size distribution changes during the modification process [24,52]. This phenomenon is even stronger when considering the mechanical/mechano-chemical modification as it is often performed in milling devices of different kinds [62,63].

In the presented research two approaches of cellulose fibre chemical modification were carried out: solvent-involving (in a flask) and non-solvent (in the planetary mill). Both treatments were associated with the possibility of UFC100 size decrease and particle size distribution variations, especially regarding the modification performed in the planetary mill.

Therefore, the hydrodynamic radii of the analysed cellulose samples dispersed in the distilled water were established. This broadened the knowledge about the possible size of the fibres before and after the performed modification process.

On the basis of the data gathered in Figure 4a, it could be observed that all of the modification processes led to a decrease of particle size. Yet, depending on the approach employed in the treatment (solvent-involving or non-solvent) some differences could be seen.

During the solvent-involving treatment, cellulose is subjected to elevated temperatures, constant mixing, and solvent presence but it is the basic environment which is believed to be the main reason for the cellulose fibre size decrease; the cleavage of some chemical bonds and hydrogen-bonding reorganization [64]. This phenomenon has been widely described in the literature [23,65]. On the other hand, during the non-solvent modification, the key factor is the mechanical degradation assigned to the milling process [52].

In general, mechano-chemical non-solvent treatment caused a more significant decrease in average particle size in comparison with a solvent-involving treatment, providing a more defined size distribution at the same time. This is not surprising regarding the fact that during the mechano-chemical treatment two processes are ongoing simultaneously: chemical modification of the cellulose fibres with a silane coupling agent and milling of UFC100 which lead to fragmentation of the particles [66]. Some process of natural fibre mechanical degradation occurs [67].

Another interesting observation is the momentous size decrease for VTMS modification of cellulose fibres. No matter which approach of UFC100 treatments is employed, the particle hydrodynamic radius decreases. Moreover, according to Figure 4g–h the cellulose size distribution is different than in the case of the rest of the modifications—a group of particles of a size below 500 nm is created. This phenomenon is not evidenced for the rest of treatments to this extent.

Furthermore, analysing the differences in particle size distribution between the modifications with the same silane coupling agent through solvent-involving and non-solvent approaches (Figure 4c–h) the following conclusion can be made—the milling process leads to the elimination of particles bigger than 2000 nm in the case of each treatment.

Yet, depending on the size of the alkyl chain in the modifying compound, the cellulose size varies, the highest being for TEOS grafting (eight carbon atoms in a chain) and the lowest for VTMS (two carbon atoms in a chain).

Taking into consideration the gathered data, non-solvent treatment is more efficient regarding particle size minimization in comparison with the solvent-involving approach. Moreover, the size of the modifying agent molecule is crucial for the hydrodynamic radius of a particle—the more carbon atoms in the silane coupling agent, the bigger the cellulose fibre particle hydrodynamic radius.

Considering polymer composite applications, the smaller filler particle size, the more efficient the tension transfer within a material may become and an improved mechanical performance observed [68]. Moreover, fibre size decrease may also affect different composite properties, e.g., transparency [69], barrier properties [70]. Yet, one should remember that the fibre size decrease leads also to some changes in the aspect ratio of the filler which is a crucial aspect regarding cellulose-based polymer composite materials [63].

### 3.3. Differential Scanning Calorimetry Analysis

Differential scanning calorimetry was employed in order to obtain more information concerning the moisture content in the analysed cellulose fibres. By investigating the enthalpy change value assigned to the water evaporation process details about the cellulose moisture absorption ability before and after the modification could be gathered.

Moisture evolution from cellulose may be described by two paths: physical—through desorption—and chemical—through elimination reactions. Moreover, the process mentioned above may be divided into three distinct temperature regimes: (i) loss of absorbed water at low temperatures (< 220 °C), (ii) loss of chemical water at moderate-to-high temperatures (220–550 °C), and (iii) loss of chemical water at high temperatures (> 550 °C) [71]. This study concentrates on the first step of moisture desorption—the physical stage.

Table 3 presents the data obtained from the conducted experiment. Moreover, Figure 5 reveals the shape of the DSC curves for different samples analysed in this research—the water evaporation process is visible (endothermic peak around 100 °C).

On the basis of data gathered in Table 3 and Figure 5, it could be observed how the enthalpy change value assigned to water desorption changes considering the different cellulose modification approaches. In general, ΔH is higher for fibres modified via a solvent-involving method (enthalpy values varies from 30.5–79.7 J/g) and decreases in the case of cellulose hydrophobized in the planetary mill (enthalpy values varies from 28.9–52.9 J/g).

Furthermore, the shift of T_peak_ towards lower temperatures for cellulose fibres modified with VTMS (particles of the lowest hydrodynamic radii) might be evidenced. Yet, considering the ΔH of these two samples, there is no correlation. This could be caused by the different modification yield for solvent-involving and non-solvent treatments.

Higher enthalpy change value is understood in the way that more energy is required to desorb the water from the cellulose surface (dehydration heat is elevated for the specimens of higher moisture content) [72]. As well, higher enthalpy change value could be described as an outcome of strong plasticizer (water)–polymer chain interactions [73]. This leads to the conclusion that some of the performed modification processes resulted in a higher moisture content in comparison with the reference sample.

On the other hand, both specimens treated with TMPS exhibit low values of ΔH. This may support the previously referred to fact that the linear silane compounds with methoxyl groups are the most suitable for cellulose modification [26]. A slightly different situation may be observed for VTMS treatment which also possess methoxyl moieties. Yet, TMPS has within its chemical structure a longer alkyl chain than VTMS, which may lead to a higher surface hydrophobisation degree.

Therefore, modifying the agent structure may play an important role in moisture content minimization, e.g., TEOS possesses an ethoxyl and not a methoxyl moiety, therefore reflecting relatively high ΔH value. The silane agent chemical structure could be a reason for its potentially lower reactivity. Consequently, moisture is more easily adsorbed on the cellulose surface.

In general, cellulose fibres modified via a mechano-chemical approach exhibit lower moisture content level. Moreover, TMPS seems to be an optimal modifying agent for cellulose fibres—TMPS grafted specimens exhibit relatively low water content and the results are reproducible considering two of the employed modification approaches. Considering the analysed data, it may be expected that TMPS leads to efficient cellulose surface hydrophobisation.

### 3.4. Correlation between the Obtained Results

Analysis of the results gathered in this research enabled observance of some possible trends upon the modified cellulose fibres concerning the structure of the modifying agent, particle size distribution, and the water evaporation process characteristics.

It is a well-known fact that the cellulose particle size has a huge impact on its water absorption and desorption ability. During the modification process, the length to diameter ratio and the specific surface area may change and, therefore, facilitate the moisture desorption process [74]. In Figure 6 a confirmation of this phenomenon may be observed. While the cellulose hydrodynamic radius increases, the temperature of the water evaporation peak visible in the DSC curve also increases. Consequently, it could be concluded that the moisture desorption process begins at a lower temperatures for cellulose fibres which are smaller.

As is visible in Figure 6 the lower the particle size, the easier it is to desorb water from the fibre surface. Nonetheless, it does not mean that this cellulose is not capable of moisture adsorption. This is a question of the modifying agent structure and its reactivity with the hydroxyl groups present on the natural fibre surface [25,26]. This dependence is shown in Figure 7a. Blocking of the hydroxyl groups on the cellulose surface by the coupling agent limits the possibilities of water adsorption. Therefore, a decreased moisture content could be observed [24,47,75].

Furthermore, as is visible in Figure 7b, generally, the bigger the chain functionalising the silane coupling agent, the higher the hydrodynamic radius of a modified cellulose sample. Therefore, smaller particles are observed in the case of UFC100 treated with VTMS and the biggest for TEOS grafting.

## 4. Conclusions

In this research a successful cellulose treatment via two different approaches was performed. Moreover, it was confirmed that the natural fibre size had a significant impact on the water evaporation temperature (the lower the size, the more decreased was the temperature of water evaporation). The fact that the structure of the modifying agent is very important regarding efficient cellulose modification was also elucidated.

The comparison of the gathered results is presented in Table 4. It can be observed that the non-solvent modification approach performed in the planetary mill is more efficient. Not only does it provide a cellulose of an altered surface but also of a significantly decreased size.

Considering all gathered information, a new one-pot cellulose modification approach, being a solution fulfilling the green chemistry requirements and compromising the sustainable development rules, was presented in this research. Regarding the decreased cellulose size, fibres produced via this approach may find application in bio-based plastics of an increased performance and elevated degradation potential. These materials would be employed, e.g., in food packaging.

## Figures and Tables

**Figure 1 materials-13-02552-f001:**
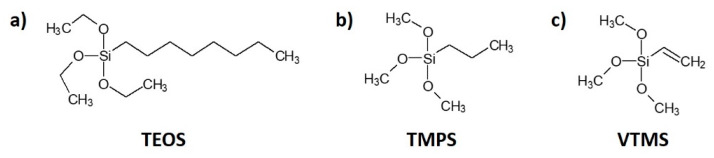
Chemical structure of silane coupling agents employed in this research: (**a**) triethoxyoctylsilane (TEOS); (**b**) trimethoxypropylsilane (TMPS); (**c**) vinyltrimethoxysilane (VTMS).

**Figure 2 materials-13-02552-f002:**
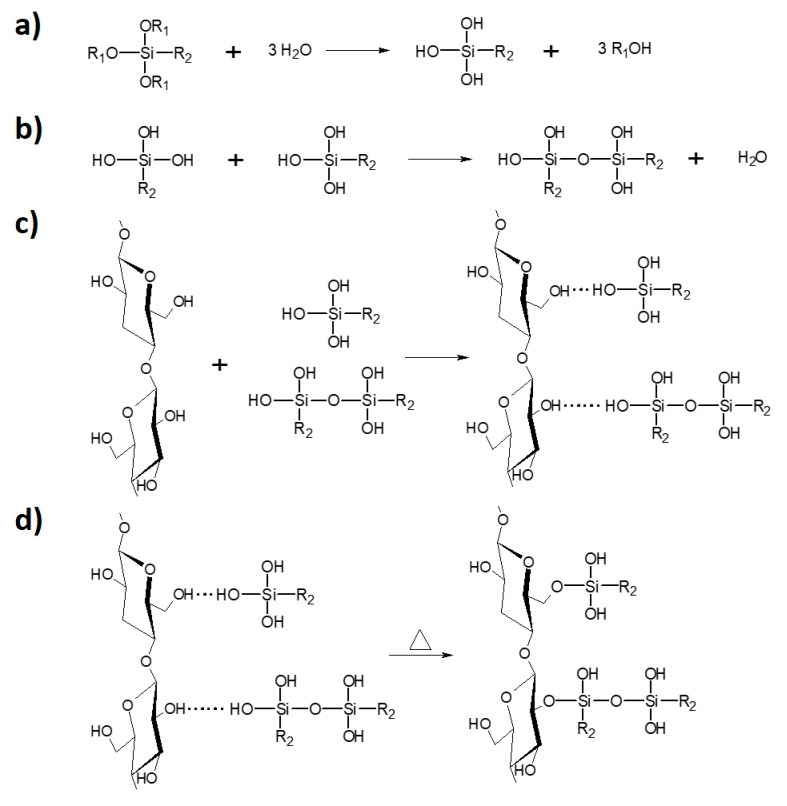
Cellulose modification with silane coupling agent: (**a**) hydrolysis; (**b**) condensation; (**c**) physical adsorption; (**d**) chemical grafting [31].

**Figure 3 materials-13-02552-f003:**
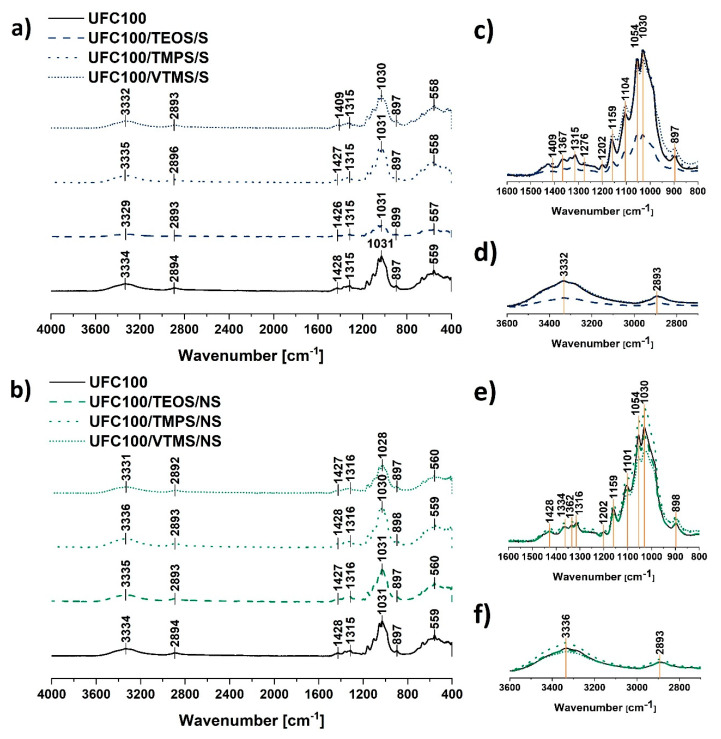
Fourier-transform infrared spectra of: (**a**) cellulose fibres modified via a solvent-involving method; (**b**) cellulose fibres modified via a non-solvent method. Characteristic absorption bands: 3334 cm^−1^ (O–H, water), 2894 cm^−1^ (C–H), 1200-900 cm^−1^ (O–H, C–O, –COO, CO–O–CO, Si–O–Si), 950–560 cm^−1^ (C–OH, C–C, Si–O–Si, Si–OH); zoom of 1600–800 cm^−1^ and 3600–2700 cm^−1^ region for spectra of solvent-involving modification (respectively: **c**,**d**) and non-solvent treatment (respectively: **e**,**f**).

**Figure 4 materials-13-02552-f004:**
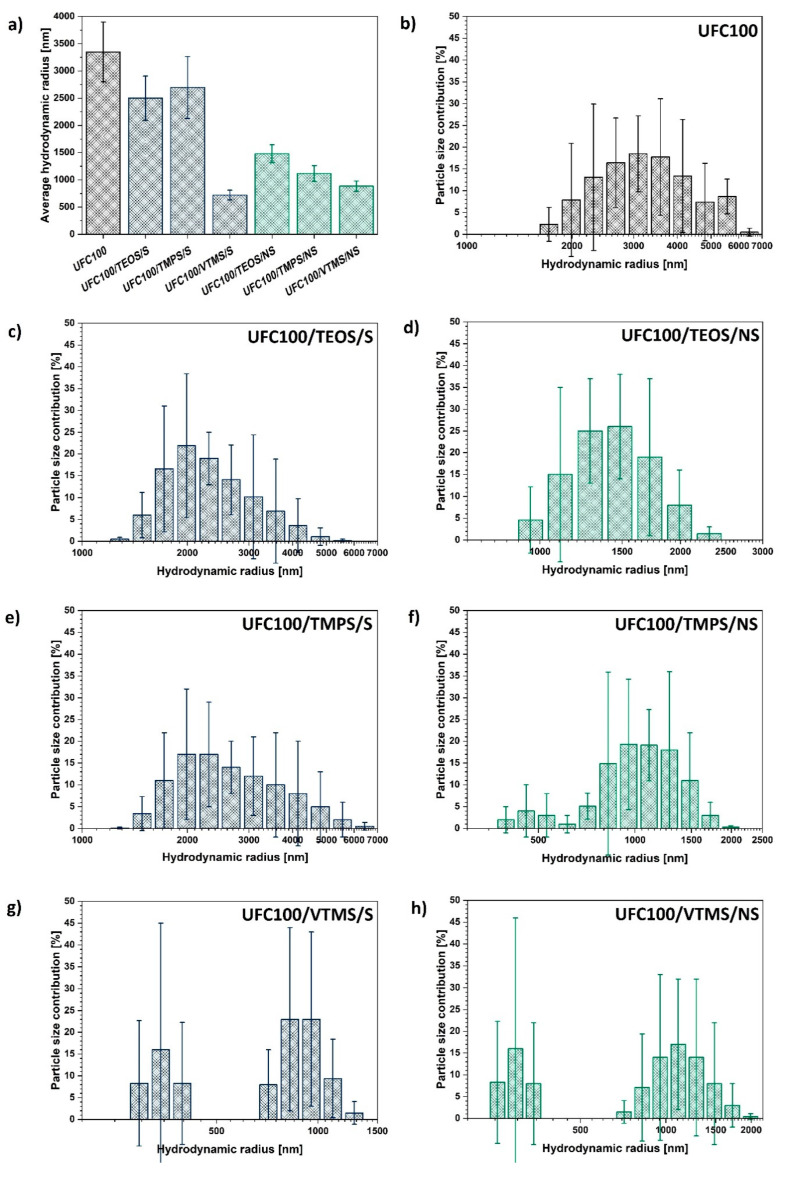
Dynamic light scattering analysis results of: (**a**) average hydrodynamic radii of analysed cellulose particles; (**b**–**h**) certain particle size distribution for different specimens.

**Figure 5 materials-13-02552-f005:**
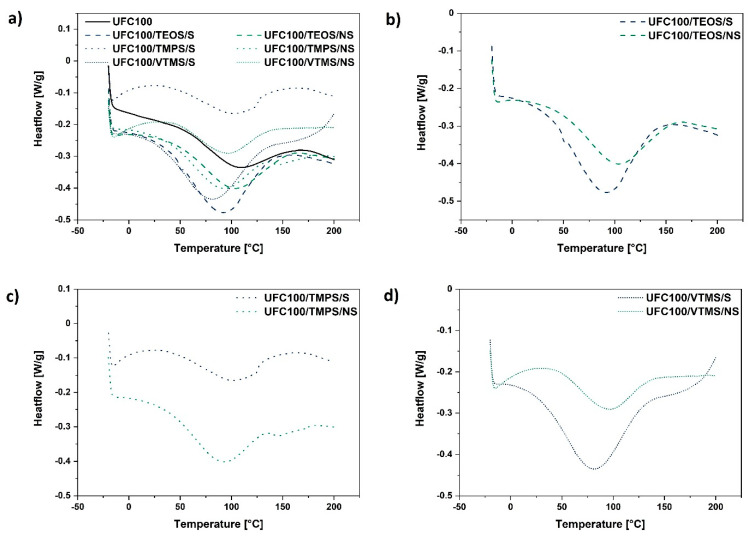
Differential scanning calorimetry curves of: (**a**) all analysed cellulose samples treated via different methods (solvent-involving/non-solvent); (**b**) cellulose specimens modified with TEOS; (**c**) cellulose specimens modified with TMPS; (**d**) cellulose specimens modified with VTMS.

**Figure 6 materials-13-02552-f006:**
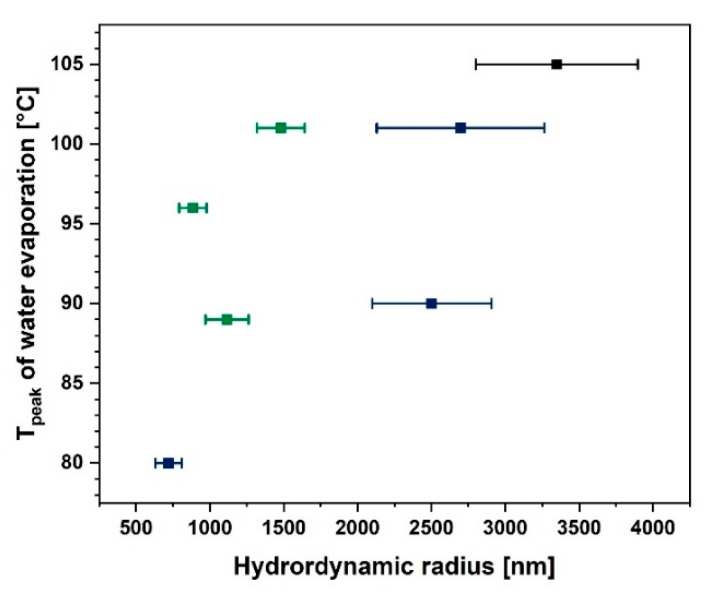
Graph illustrating water evaporation peak temperature as a function of particle hydrodynamic radius.

**Figure 7 materials-13-02552-f007:**
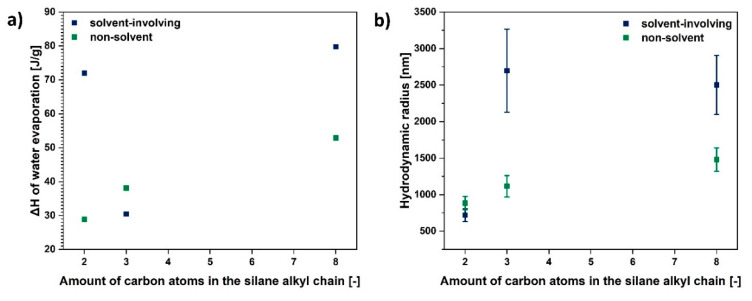
Graphs illustrating: (**a**) ΔH assigned to water evaporation and (**b**) hydrodynamic radius as a function of amount of carbon atoms in the silane alkyl chain.

**Table 1 materials-13-02552-t001:** Summary of performed cellulose modifications. Abbreviations: TEOS—triethoxyoctylsilane; TMPS—trimethoxypropylsilane; VTMS—vinyltrimethoxysilane.

Sample	Silane			Modification Type	
	TEOS	TMPS	VTMS	Solvent-Involving Approach (S)	Non-Solvent Approach (NS)
UFC100/TEOS/S	**✔**	**-----**	**-----**	**✔**	**-----**
UFC100/TMPS/S	**-----**	**✔**	**-----**	**✔**	**-----**
UFC100/VTMS/S	**-----**	**-----**	**✔**	**✔**	**-----**
UFC100/TEOS/NS	**✔**	**-----**	**-----**	**-----**	**✔**
UFC100/TMPS/NS	**-----**	**✔**	**-----**	**-----**	**✔**
UFC100/VTMS/NS	**-----**	**-----**	**✔**	**-----**	**✔**

**Table 2 materials-13-02552-t002:** Wavenumber values assigned to the chemical moieties.

Wavenumber [cm^−1^]	Chemical Group	Ref.
560	C–OH out-of-plane bending, C–C	[38]
950–750	Si–O–Si, Si–OH	[23,39]
1200–900	–OH, –COO	[35]
1100–1000	CO–O–CO	[40]
1030	C–O stretching vibration	[41]
1100	–OH, Si–O–Si	[33,42]
1150	C–O stretching vibration, C–O–C bridge	[43]
1240	–CH_3_	[44]
1300–1100	C–O, C=O, C=C, COOH	[33]
1450	C–H bending of CH_2_, Si–O–C	[23,45]
1650	OH bending of adsorbed water, C=C	[46]
2900–2800	CH stretching vibration	[34]
3330	–OH, water	[32]

**Table 3 materials-13-02552-t003:** Tabularized water evaporation peak temperatures (T_peak_) and the assigned enthalpy values (ΔH) to this phenomenon for all analysed specimens.

Sample	T_peak_ [°C]	ΔH [J/g]
UFC100	105	41.1
UFC100/TEOS/S	90	79.7
UFC100/TMPS/S	101	30.5
UFC100/VTMS/S	80	72.0
UFC100/TEOS/NS	101	52.9
UFC100/TMPS/NS	89	38.1
UFC100/VTMS/NS	96	28.9

**Table 4 materials-13-02552-t004:** Comparison of solvent-involving (S) and non-solvent (NS) modification approaches.

Comparison	Solvent-Involving Approach (S)	Non-Solvent Approach (NS)
Reaction time [h]	2	2
Steps	1. chemical modification in a flask 2. solvent distillation 3. silane coupling in a dryer	1. mechano-chemical modification in a planetary mill vessel 2. silane coupling in a dryer
Additional waste	solvent	--------
**Treatment Effect on the Cellulose Properties**		
FT-IR/NIR	shifts between the absorption bands, possible degradation of glucose rings	stronger shifts between the absorption bands, possible degradation of glucose rings
DLS	particle size decrease	more significant particle size decrease, particles bigger than 2000 nm elimination
DSC	higher moisture content	lower moisture content, water evaporation at decreased temperatures

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
