# Peer review of "Superiority of Cellulose Non-Solvent Chemical Modification over Solvent-Involving Treatment: Solution for Green Chemistry (Part I)"

_materials, 2020, doi:10.3390/ma13112552_

Round 1
Reviewer 1 Report
From the standpoint of minimizing the environmental impacts of industrial processing, it is often assumed that adverse impacts can be minimized by avoiding the use of solvents. Some solvents are toxic, and volatile compounds emitted to the air can cause various harm or increase the energy required to complete the operation. But on the other hand, solvents can facilitate processing, and in many cases they can be recovered and used multiple times. The authors have carried out one of the few studies that have compared solvent-based and solvent-free procedures for modification of the surface of cellulosic materials. Planetary mill treatment represents a meaningful step in the direction of practical industrial processing. The work was well justified in the Introduction.
- Introduction: A general introduction to factors affecting the life-cycle impacts of various treatments to modify cellulose surfaces, including silane-based treatments, appears in the following review article: Hubbe, M. A., Rojas, O. J., and Lucia, L. A. (2015). "Green modification of surface characteristics of cellulosic materials at the molecular or nano scale: A review," BioResources 10(3), 6095-6229. DOI: 10.15376/biores.10.3.Hubbe
- Line 98 and following lines: What was the moisture content of the cellulose? This is an especially critical factor with respect to reaction with silane chemicals, both under dry conditions and in solvent. Future researchers will not be able to faithfully repeat the work and get the same results without knowing the moisture content of the cellulose that was used in the work. For instance, in Fig. 2, part a, H2O is part of the reaction scheme. Also, the authors need to estimate the amount of water present in the solvent.
- A key aspect appears to have been overlooked by the authors and, in my opinion, it is really needed before the work can be regarded as complete: The whole point of treatment of cellulosic surfaces with a trialkoxysilane derivative usually is to change the wettability characteristics of the surfaces. It is relatively quick and easy to obtain contact information on cellulosic surfaces. The authors need to compare the water contact angles before and after each kind of treatment. More sophisticated analyses of wettability are also possible, but a simple comparison of water contact angles would be the minimum that readers would expect. For future work it would be important to show whether or not the modified cellulose particles would have different performance as the reinforcement of plastic composites (also include the untreated cellulose as a control).
- The authors spent a lot of effort and a lot of text and figures in an attempt to determine whether or not silane-related chemicals had been retained and grafted. Due to the nature of FTIR, the degree of success in this effort is completely understandable. However, it is important to keep in mind that the extent of retention and the extent of reaction are not so critically important relative to the main goal of the work, which is to determine the environmental impacts of the two main preparation strategies. So it would be an acceptable option for the authors to reduce the amount of text pertaining to FTIR and to skip some of the figures or place them into a “supplementary materials” section.
Author Response
Institute of Polymer and Dye Technology
Technical University of Lodz
90-924 Lodz, ul Stefanowskiego 12/16, Poland
Tel.: +48 42 631 32 23, Fax: +48 42 636 25 43
May 15, 2020
Materials
Dear Professor,
We are resubmitting our revised paper entitled Superiority of cellulose non-solvent chemical modification over solvent-involving treatment: solution for green chemistry (part I) by, Stefan Cichosz, Anna Masek with a request to reconsider it for publication in Materials.
We have carefully considered the Editor and Reviewers' comments. The manuscript was revised exactly according to these comments. The list of responses to the reviewer’s comments and corrections made in the manuscript is attached.
The manuscript has not been previously published, is not currently submitted for review to any other journal, and will not be submitted elsewhere before a decision is made by this journal.
For correspondence please use the following information:
corresponding author: Anna Masek
Institute of Polymer and Dye Technology
Technical University of Lodz
90-924 Lodz, ul Stefanowskiego 12/16, Poland
Tel.: +48 42 631 32 93
Fax: +48 42 636 25 43
e-mail: anna.masek@p.lodz.pl
Yours sincerely,
Ph. D., D.Sc. Anna Masek
All changes are marked with a green colour through whole manuscript.
Reviewer #1
From the standpoint of minimizing the environmental impacts of industrial processing, it is often assumed that adverse impacts can be minimized by avoiding the use of solvents. Some solvents are toxic, and volatile compounds emitted to the air can cause various harm or increase the energy required to complete the operation. But on the other hand, solvents can facilitate processing, and in many cases they can be recovered and used multiple times. The authors have carried out one of the few studies that have compared solvent-based and solvent-free procedures for modification of the surface of cellulosic materials. Planetary mill treatment represents a meaningful step in the direction of practical industrial processing. The work was well justified in the Introduction.
The comments are listed below:
- Introduction: A general introduction to factors affecting the life-cycle impacts of various treatments to modify cellulose surfaces, including silane-based treatments, appears in the following review article: Hubbe, M. A., Rojas, O. J., and Lucia, L. A. (2015). "Green modification of surface characteristics of cellulosic materials at the molecular or nano scale: A review," BioResources 10(3), 6095-6229. DOI: 10.15376/biores.10.3.Hubbe
Answer: We are grateful for enriching our references with such a well-written article which is precise in description of cellulose silanisation methods. The reference has been added: Silanes may affect different properties of cellulose, e.g., moisture content, thermal decomposition, surface characteristics, fibre size [21–25].
- Line 98 and following lines: What was the moisture content of the cellulose? This is an especially critical factor with respect to reaction with silane chemicals, both under dry conditions and in solvent. Future researchers will not be able to faithfully repeat the work and get the same results without knowing the moisture content of the cellulose that was used in the work. For instance, in Fig. 2, part a, H2O is part of the reaction scheme. Also, the authors need to estimate the amount of water present in the solvent.
Answer: We are thankful for drawing our attention to this problem. We will remember about this aspect in future. Fortunately, the moisture content in the investigated cellulose fibres was a subjects of our previous research which included similar conditions: S. Cichosz, A. Masek, Drying of the natural fibers as a solvent-free way to improve the cellulose-filled polymer composite performance, Polymers, 2020, 12, 484-502. doi: 10.3390/polym12020484. Therefore, the following information has been added to the manuscript:
- Cellulose moisture content of approximately 6wt% (estimated on the basis of the previous study [28]).
- Ethanol (concentration: 96%) employed as a reaction environment in the solvent-involving modification approach has been purchased from Chempur (…)
Unfortunately, the bottle with ethanol employed in this research is over and we are unable to estimate the moisture content of this exact product. Therefore,
the concentration information was added.
- A key aspect appears to have been overlooked by the authors and, in my opinion, it is really needed before the work can be regarded as complete: The whole point of treatment of cellulosic surfaces with a trialkoxysilane derivative usually is to change the wettability characteristics of the surfaces. It is relatively quick and easy to obtain contact information on cellulosic surfaces.
Answer: We agree that the contact angle information is important regarding a trialkoxysilane derivative treatment of natural fibres. We did not have investigated the wettability of the cellulose fibres due to the fact that the filler-polymer matrix interface properties have been described in the Part II of this article. Nowadays, we are unable to carry out this experiment due to the lockdown. Yet, we will remember about this key aspect during the future work. We hope that considering the current pandemic situation it will not be a problem. We are thankful for this enriching and valuable comment.
- The authors need to compare the water contact angles before and after each kind of treatment. More sophisticated analyses of wettability are also possible, but a simple comparison of water contact angles would be the minimum that readers would expect. For future work it would be important to show whether or not the modified cellulose particles would have different performance as the reinforcement of plastic composites (also include the untreated cellulose as a control). The authors spent a lot of effort and a lot of text and figures in an attempt to determine whether or not silane-related chemicals had been retained and grafted. Due to the nature of FTIR, the degree of success in this effort is completely understandable. However, it is important to keep in mind that the extent of retention and the extent of reaction are not so critically important relative to the main goal of the work, which is to determine the environmental impacts of the two main preparation strategies. So it would be an acceptable option for the authors to reduce the amount of text pertaining to FTIR and to skip some of the figures or place them into a “supplementary materials” section.
Answer: We are grateful for this comment. We agree that it was our mistake to put so much stress on the FT-IR investigation. As a consequence, NIR analysis has been moved to the Supplementary materials section. Moreover, we think that the Reviewer’s comments were incredibly valuable and we would improve our study in the future with this knowledge.
Reviewer 2 Report
The article submmited by Cichosz and Mased entitled “Superiority of cellulose non-solvent chemical mofication over solvent-involving treatment: solution for green chemistry (part I) is about the cellulose modification via non-solvent involving treatment. The modification has been analyzed using several characterization technique as FITR, DLS, DSC.
Abstract
Not relevant information should be eliminated. Line 16 – 17: “the particles bigger than 2000 nm have been eliminated.”. It is explained in the manuscrit.
Specific data of the more relevant results should be added into the abstract.
Introduction
Line 37 – 38: Authors expose that the surface hydrophobization of cellulose is recommended to improve the interface adhesion between fiber and non-polar polymer matrix. There are other techniques also described in literature as the use of compatibilizers such as MAPE, MAPP, etc. Add it.
Line 44: What is Eucaliposis binata? Is it a plant specie? If it is, name it properly.
Results
Section 3.1.: In FTIR technique no differences were observed between treanted and untreated fiber. Is it neccesary to keep it in the article? The authors should link the no difference observation of this technique with the need of realize the other techniques.
Line 205 – 209: This statement is impossible to prove. The authors should remove it or justify better.
Line 275: Describe the degradation mechanism
Line 279: in “the” case
Line 291-294: Why the authors define the reduction of the size fiber as an advantage? It can affect to the aspect ratio of fiber that is a key parameter in the use of natural finber in polymer reinforcement.
Author Response
Institute of Polymer and Dye Technology
Technical University of Lodz
90-924 Lodz, ul Stefanowskiego 12/16, Poland
Tel.: +48 42 631 32 23, Fax: +48 42 636 25 43
May 15, 2020
Materials
Dear Professor,
We are resubmitting our revised paper entitled Superiority of cellulose non-solvent chemical modification over solvent-involving treatment: solution for green chemistry (part I) by, Stefan Cichosz, Anna Masek with a request to reconsider it for publication in Materials.
We have carefully considered the Editor and Reviewers' comments. The manuscript was revised exactly according to these comments. The list of responses to the reviewer’s comments and corrections made in the manuscript is attached.
The manuscript has not been previously published, is not currently submitted for review to any other journal, and will not be submitted elsewhere before a decision is made by this journal.
For correspondence please use the following information:
corresponding author: Anna Masek
Institute of Polymer and Dye Technology
Technical University of Lodz
90-924 Lodz, ul Stefanowskiego 12/16, Poland
Tel.: +48 42 631 32 93
Fax: +48 42 636 25 43
e-mail: anna.masek@p.lodz.pl
Yours sincerely,
Ph. D., D.Sc. Anna Masek
Reviewer #2
The article submitted by Cichosz and Masek entitled “Superiority of cellulose non-solvent chemical modification over solvent-involving treatment: solution for green chemistry (part I)” is about the cellulose modification via non-solvent involving treatment. The modification has been analyzed using several characterization technique as FITR, DLS, DSC.
The comments are listed below:
- Abstract. Not relevant information should be eliminated. Line 16 – 17: “the particles bigger than 2000 nm have been eliminated.”. It is explained in the manuscrit. Specific data of the more relevant results should be added into the abstract.
Answer: We have corrected this part on the Reviewer’s request: This provides the opportunity of cellulose mechanical degradation, decreasing its size, simultaneously with the ongoing silane coupling agent grafting. FT-IR indicated the possibility of the intense cleavage of glucose rings in cellulose chains during the mechano-chemical treatment. This effect has been proved with DLS results – the size of the particles decreased.
- Introduction. Line 37 – 38: Authors expose that the surface hydrophobization of cellulose is recommended to improve the interface adhesion between fiber and non-polar polymer matrix. There are other techniques also described in literature as the use of compatibilizers such as MAPE, MAPP, etc. Add it. Line 44: What is Eucaliposis binata? Is it a plant species? If it is, name it properly.
Answer: We have never claimed that the silanisation is the only method for reinforcing cellulose-based polymer composites: Cellulose is often modified with silanes as they are substances of a high efficiency considering the hydrophobisation of natural fibre surface [2,19,20]. Nevertheless, we may understand that this statement is misleading. Therefore, we have broadened this section with further information: There are known different methods of natural fibre modification, e.g., employing maleinized polymer matrix as a compatibilizer [19,20]. We have also corrected the typo in the plant name: Eulaliopsis binata. We are thankful for this comment.
- Results. Section 3.1.: In FTIR technique no differences were observed between treated and untreated fiber. Is it neccesary to keep it in the article? The authors should link the no difference observation of this technique with the need of realize the other techniques. Line 205 – 209: This statement is impossible to prove. The authors should remove it or justify better. Line 275: Describe the degradation mechanism Line 279: in “the” case Line 291-294: Why the authors define the reduction of the size fiber as an advantage? It can affect to the aspect ratio of fiber that is a key parameter in the use of natural fiber in polymer reinforcement.
Answer: According to Reviewer’s #1 comments, NIR analysis has been moved to the Supplementary materials section. Yet, we do not agree that there are no visible changes in the FTIR spectra. We have improved our description of the results. Following information was added:
- As it is impossible to distinguish the intensity variations. It is advised to analyse the shifts between the absorption bands which is more accurate method [46,48,49].
- Interestingly, regarding main absorption bands visible in Fig. 3, shifts between the peaks could be detected. It is crucial considering the possibility of different interaction characteristics among cellulose fibres caused by some structural variations. Visible changes are stronger in case of non-solvent treatment. This indicates that the fibres modified with this method are altered more during the carried out process. Many researchers, explain absorption band shifts as factors indicating significant structural changes within cellulosic materials. Furthermore, the shifts are considered to be more accurate than peak intensity changes [46,48–51].
- During the solvent-involving treatment, cellulose is subjected to elevated temperatures, constant mixing and solvent presence but it is the basic environment which is believed to be a main reason of the cellulose fibre size decrease, cleavage of some chemical bonds and hydrogen-bonding reorganization [64]. This phenomenon has been widely described in literature [23,65]. On the other hand, during the non-solvent modification, the key factor is mechanical degradation assigned to the milling process [52].
- Considering polymer composite applications, the smaller filler particle size, the more efficient tension transfer within a material may become and the improved mechanical performance observed [68]. Moreover, fibre size decrease may also affect different composite properties, e.g., transparency [69], barrier properties [70]. Yet, one should remember that the fibre size decrease leads also to some changes in the aspect ratio of the filler which is a crucial aspect regarding cellulose-based polymer composite materials [63].
Reviewer 3 Report
The submitted manuscript deals with a comparison of non solvent Vs solvent assisted cellulose silanization. While I found the topic developed interesting I’m afraid this manuscript should be rejected at present stage due to the following reasons:
- The overall quality of language must be improved, some sentences might not deliver the original message the authors write them for (for instance page 2 line 83)
- The performed IR characterization is rather limited, the way the spectra are presented makes a comparison impossible. The authors should provide zoomed in sections where Si-O signals are likely to be found and evaluate potential differences from unmodified cellulose.
- DLS is employed but achieved data show large errors making a comparison between samples very difficult and limited.
- DSC is used to evaluate moisture after drying at 100°C for 24h , were the samples dried to constant weight ? have the authors considered to stabilize the samples in controlled conditions ( e.g. 23°C 50% RH) or performing TGA ?
- DSC plots show rather different baselines that could possibly affect the value of the calculated enthalpy values
Author Response
Institute of Polymer and Dye Technology
Technical University of Lodz
90-924 Lodz, ul Stefanowskiego 12/16, Poland
Tel.: +48 42 631 32 23, Fax: +48 42 636 25 43
May 15, 2020
Materials
Dear Professor,
We are resubmitting our revised paper entitled Superiority of cellulose non-solvent chemical modification over solvent-involving treatment: solution for green chemistry (part I) by, Stefan Cichosz, Anna Masek with a request to reconsider it for publication in Materials.
We have carefully considered the Editor and Reviewers' comments. The manuscript was revised exactly according to these comments. The list of responses to the reviewer’s comments and corrections made in the manuscript is attached.
The manuscript has not been previously published, is not currently submitted for review to any other journal, and will not be submitted elsewhere before a decision is made by this journal.
For correspondence please use the following information:
corresponding author: Anna Masek
Institute of Polymer and Dye Technology
Technical University of Lodz
90-924 Lodz, ul Stefanowskiego 12/16, Poland
Tel.: +48 42 631 32 93
Fax: +48 42 636 25 43
e-mail: anna.masek@p.lodz.pl
Yours sincerely,
Ph. D., D.Sc. Anna Masek
Reviewer #3
The submitted manuscript deals with a comparison of non-solvent Vs solvent assisted cellulose silanization. While I found the topic developed interesting I’m afraid this manuscript should be rejected at present stage due to the following reasons.
The comments are listed below:
- The overall quality of language must be improved, some sentences might not deliver the original message the authors write them for (for instance page 2 line 83).
Answer: We are thankful for this comment. The whole manuscript has been revised carefully considering some language mistakes and appropriate meaning. Changes have been marked throughout the manuscript.
- The performed IR characterization is rather limited, the way the spectra are presented makes a comparison impossible. The authors should provide zoomed in sections where Si-O signals are likely to be found and evaluate potential differences from unmodified cellulose.
Answer: We are grateful for this advice. Therefore, the Figure presenting FT-IR spectra of analysed samples has been improved and some more description added:
- FT-IR spectra of UFC100 (Fig. 3c-f) exhibit some characteristic absorption bands, e.g., hydroxyl moieties at 3334 cm-1 [32] and 1030 cm-1 [33], C-H stretching vibration at 2896 cm-1 [34], -COO at 1200-900 cm-1 [35] which were also detected in different works [36,37]. Absorption bands assigned to the exact moieties has been presented in Table 2 (…) Only UFC100/TEOS/S exhibit some differences in comparison with the reference sample. Nevertheless, it is only a question of the whole spectrum intensity – it is lower. Repeated FT-IR measurements have resulted in obtaining the spectrum of the same intensity (…) The peaks which correspond to the C-O, C-C stretching vibrations and CH2 rocking vibration at 1100-900 cm-1 [35,43] are also assigned to the Si-O-Si (1100 cm-1, 950-800 cm-1), Si-O-C (1450 cm-1), Si-OH (900 cm-1), and C-Si-C bonds (663 cm-1) [23,39,42]. The phenomenon of overlapping for the silanization of the cellulose fibres was also observed in previous studies [47]. Fig. 3c and Fig. 3e reveal how difficult it is to evidence some specific variations which could confirm silane grafting. Changes in the intensities of these peaks may indicate some information about the, e.g., cellulose degradation, as well as they might reveal silane coupling agent grafting on the fibre surface. Nevertheless, taking into consideration that these two processes are ongoing simultaneously, it is impossible to distinguish the intensity variations (…) Likewise, new absorption bands do not appear in the FT-IR spectra, as silane coupling agents employed in this research are functionalised with alkyl chains of different length and structure – new functional groups are not introduced into the system; only silicon atom containing moieties and carbon-based chains. Consequently, one should consider some subtle variations in the FT-IR spectra in order to confirm the modification process occurrence. As it is impossible to distinguish the intensity variations. It is advised to analyse the shifts between the absorption bands which is more accurate method [46,48,49]. Interestingly, regarding main absorption bands visible in Fig. 3, shifts between the peaks could be detected. It is crucial considering the possibility of different interaction characteristics among cellulose fibres caused by some structural variations. Visible changes are stronger in case of non-solvent treatment. This indicates that the fibres modified with this method are altered more during the carried out process. Many researchers, explain absorption band shifts as factors indicating significant structural changes within cellulosic materials. Furthermore, the shifts are considered to be more accurate than peak intensity changes [46,48–51].
- DLS is employed but achieved data show large errors making a comparison between samples very difficult and limited.
Answer: We agree with this comment. The error bars regarding the size distribution are huge. Nevertheless, we think some trends between the samples are visible. Moreover, error bars considering the average values of hydrodynamic radii are not so huge. This is only a problem of size distribution.
- DSC is used to evaluate moisture after drying at 100°C for 24h , were the samples dried to constant weight ? have the authors considered to stabilize the samples in controlled conditions ( e.g. 23°C 50% RH) or performing TGA?
Answer: Yes, samples were dried to constant weight. We have tested it before. The accurate measurements considering the moisture content in cellulose fibres have been a subject of our previous research: S. Cichosz, A. Masek, Drying of the natural fibers as a solvent-free way to improve the cellulose-filled polymer composite performance, Polymers, 2020, 12, 484-502. doi: 10.3390/polym12020484. We believe that in order to estimate the moisture content in the analysed samples various scientific techniques should be employed at the same time. The aim of this article was not to investigate only the moisture content, but the overall variations in cellulose properties after the solvent-involving and the non-solvent treatment. Now, we are carrying out another research study considering the cellulose conditioning in various conditions and testing its properties regarding the moisture content.
- DSC plots show rather different baselines that could possibly affect the value of the calculated enthalpy values.
Answer: We believe that calculated enthalpy values were well-calculated regarding different baselines. We attach the following calculation reports below (Fig. 1-6):
Fig. 1 DSC curve of UFC100/VTMS/NS.
Fig. 2 DSC curve of UFC100/TMPS/NS.
Fig. 3 DSC curve of UFC100/TEOS/NS.
Fig. 4 DSC curve of UFC100/VTMS/S.
Fig. 5 DSC curve of UFC100/TMPS/S.
Fig. 6 DSC curve of UFC100/TEOS/S.
Round 2
Reviewer 1 Report
I noticed that the authors failed to copy the actual title of a cited article, just using an abbreviated form of the title. The authors gave the reference information for item # 27 as follows: Hubbe, M.A.; Rojas, O.J.; Lucia, L.A. Surface modification: Review. BioResources 2015, 10, 6095–6206, The correct citation, with the full title, is as follows: Hubbe, M.A.; Rojas, O.J.; Lucia, L.A. Green modification of surface characteristics of cellulosic materials at the molecular or nano scale: A review. BioResources 2015, 10, 6095–6206
Apparently the authors copied the abbreviated version of the title from the footer of the article.
Author Response
Institute of Polymer and Dye Technology
Technical University of Lodz
90-924 Lodz, ul Stefanowskiego 12/16, Poland
Tel.: +48 42 631 32 23, Fax: +48 42 636 25 43
May 26, 2020
Materials
Dear Professor,
We are resubmitting our revised paper entitled Superiority of cellulose non-solvent chemical modification over solvent-involving treatment: solution for green chemistry (part I) by, Stefan Cichosz, Anna Masek with a request to reconsider it for publication in Materials.
We have carefully considered the Editor and Reviewers' comments. The manuscript was revised exactly according to these comments. The list of responses to the reviewer’s comments and corrections made in the manuscript is attached.
The manuscript has not been previously published, is not currently submitted for review to any other journal, and will not be submitted elsewhere before a decision is made by this journal.
For correspondence please use the following information:
corresponding author: Anna Masek
Institute of Polymer and Dye Technology
Technical University of Lodz
90-924 Lodz, ul Stefanowskiego 12/16, Poland
Tel.: +48 42 631 32 93
Fax: +48 42 636 25 43
e-mail: anna.masek@p.lodz.pl
Yours sincerely,
Ph. D., D.Sc. Anna Masek
All changes are marked with a green colour through whole manuscript.
Reviewer #1
The comments are listed below:
- I noticed that the authors failed to copy the actual title of a cited article, just using an abbreviated form of the title. The authors gave the reference information for item # 27 as follows: Hubbe, M.A.; Rojas, O.J.; Lucia, L.A. Surface modification: Review. BioResources 2015, 10, 6095–6206, The correct citation, with the full title, is as follows: Hubbe, M.A.; Rojas, O.J.; Lucia, L.A. Green modification of surface characteristics of cellulosic materials at the molecular or nano scale: A review. BioResources 2015, 10, 6095–6206 Apparently the authors copied the abbreviated version of the title from the footer of the article.
Answer: We are thankful for this comment. The mentioned reference was corrected: Hubbe, M.A.; Rojas, O.J.; Lucia, L.A. Green modification of surface characteristics of cellulosic materials at the molecular or nano scale: A review. BioResources 2015, 10, 6095–6206.
Reviewer 2 Report
The changes has been made. In my opinion the article is ready to be published.
Author Response
Institute of Polymer and Dye Technology
Technical University of Lodz
90-924 Lodz, ul Stefanowskiego 12/16, Poland
Tel.: +48 42 631 32 23, Fax: +48 42 636 25 43
May 26, 2020
Materials
Dear Professor,
We are resubmitting our revised paper entitled Superiority of cellulose non-solvent chemical modification over solvent-involving treatment: solution for green chemistry (part I) by, Stefan Cichosz, Anna Masek with a request to reconsider it for publication in Materials.
We have carefully considered the Editor and Reviewers' comments. The manuscript was revised exactly according to these comments. The list of responses to the reviewer’s comments and corrections made in the manuscript is attached.
The manuscript has not been previously published, is not currently submitted for review to any other journal, and will not be submitted elsewhere before a decision is made by this journal.
For correspondence please use the following information:
corresponding author: Anna Masek
Institute of Polymer and Dye Technology
Technical University of Lodz
90-924 Lodz, ul Stefanowskiego 12/16, Poland
Tel.: +48 42 631 32 93
Fax: +48 42 636 25 43
e-mail: anna.masek@p.lodz.pl
Yours sincerely,
Ph. D., D.Sc. Anna Masek
All changes are marked with a green colour through whole manuscript.
Reviewer #2
The changes has been made. In my opinion the article is ready to be published.